# RECONCILING FEATURE SHARING AND MULTIPLE PREDICTIONS WITH MIMO VISION TRANSFORMERS

## ABSTRACT

Multi-input multi-output training improves network performance by optimizing multiple subnetworks simultaneously. In this paper, we propose MixViT, the first MIMO framework for vision transformers that takes advantage of ViTs' innate mechanisms to share features between subnetworks. This is in stark contrast to traditional MIMO CNNs that are limited by their inability to mutualize features. Unlike them, MixViT only separates subnetworks in the last layers thanks to a novel source attribution that ties tokens to specific subnetworks. As such, we retain the benefits of multi-output supervision while training strong features useful to both subnetworks. We verify MixViT leads to significant gains across multiple architectures (ConViT, CaiT) and datasets (CIFAR, TinyImageNet, ImageNet-100, and ImageNet-1k) by fitting multiple subnetworks at the end of a base model.

## 1 INTRODUCTION

Training deep architectures has become commonplace in modern machine learning applications (Krizhevsky et al., 2012; He et al., 2016) over the course of the last decade. Finding ways to better train these models has therefore become a question of paramount importance, and thus led to significant work in the literature (Zhang et al., 2018; Shanmugam et al., 2021).

Multi-input multi-output (MIMO) architectures provide a promising (Havasi et al., 2021; Rame et al., 2021) technique to maximize model performance. These methods train 2 concurrent subnetworks within a base network by mixing 2 inputs into a single one, extracting a shared feature representation with a core network and retrieving 2 predictions, one for each input. At inference, we use the same input twice and get 2 predictions. This strongly benefits model performance both by emulating ensembling (Lakshminarayanan et al., 2017; Hansen & Salamon, 1990) with subnetwork predictions (Havasi et al., 2021) and by implementing a particular form (Rame et al., 2021; Sun et al., 2022) of mixing samples data augmentation (MSDA) (Zhang et al., 2018; Yun et al., 2019).

While one would hope the resulting subnetworks cooperate to learn and share strong generic features, this usually is not the case (Rame et al., 2021): if a feature is used by one subnetwork, the other does not use it (see the visualization given for reference at the bottom of Fig. 1). This in turn causes issues for MIMO networks like forcing the use of large base networks. Although Sun et al. (2022) shows this separation issue stems from a difficulty in reconciling the shared representations with the need for distinct predictions, this is not a trivial problem to overcome in existing MIMO CNNs.

Vision Transformers (Dosovitskiy et al., 2021; Touvron et al., 2021a) (ViT) offer a new solution to this problem with their token-based representation and attention mechanism that could address these issues. Remarkably, MIMO networks' success has until now remained confined to Convolutionnal Neural Networks and the paradigm has yet to extend to these emerging vision transformers. This is all the more notable as they have started outperforming CNNs on vision benchmarks (Touvron et al., 2021a; Liu et al., 2021b) and seem well suited to train more efficient MIMO transformers.

In this paper, we propose the first ViT based MIMO framework, MixViT, that significantly improves upon the performance of standard single-input single-output models at a minimal cost. MixViT modifies the traditional MIMO structure to take full advantage of ViTs' propensity to mutualize features between subnetworks while still retaining the advantage of training distinct predictions (see Fig. 1). It therefore avoids common MIMO issues, and even benefits from strong implicit regularization (due to feature sharing) that proves particularly useful on smaller datasets. At training,

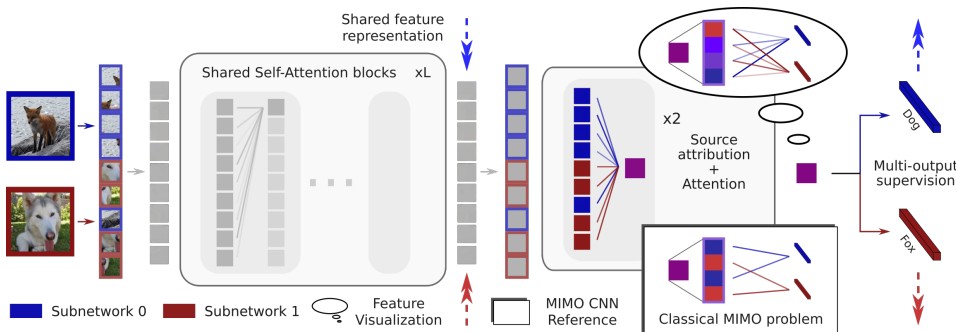

Figure 1: **MixViT trains efficient MIMO ViTs.** We aim to mutualize features between subnetworks as in the visualization bubble (instead of disjoint features like in classical MIMO) by reconciling the need to predict distinct outputs with the need for features to describe both inputs.

MixVit mixes patch tokens from 2 inputs before feeding them to shared Self-Attention blocks. To help extract 2 predictions from these shared features, we then introduce "source attribution" to associate each attended token to the input/subnetwork they pertain to and aggregate features with a class token. Finally, we get 2 predictions from the class token features with 2 different dense layers. At inference, MixViT feeds a unique input to the model and retrieves two predictions thanks to source attribution.

Our contributions are therefore as follows: **1)** We leverage innate properties of Vision Transformers to train efficient subnetworks that share features as needed. MixViT therefore adds only marginal costs at training and inference while strongly regularizing the model. **2)** We introduce a novel source attribution mechanism to facilitate this late separation of subnetworks. **3)** We propose the first working MIMO ViT as MixViT solves issue of traditional MIMO frameworks on transformers.

MixViT sets a new state-of-the-art on the TinyImageNet dataset and shows strong benefits across multiple architectures (ConViT (D'Ascoli et al., 2021), CaiT (Touvron et al., 2021b)) and datasets (CIFAR-10/100 (Krizhevsky et al., 2009), TinyImageNet (Chrabaszcz et al., 2017), ImageNet-100 (Tian et al., 2020)), and ImageNet-1k (Deng et al., 2009).

## 2  MIXVIT

We propose MixViT (Fig. 2), a MIMO framework for vision transformers that takes advantage of ViTs' innate properties to share features between subnetworks while still yielding diverse predictions. MixViT adapts the classical MIMO workflow to ViTs by completely restructuring the model so that subnetworks share early features and only specialize in the last layers thanks to our novel "source attribution" mechanism which allows the emergence of independent subnetworks in the last layers. This section starts by introducing MixViT (Sec. 2.1) before discussing MixViT's new MIMO structure in Sec. 2.2, our new source attribution (Sec. 2.3) mechanism, and its added overhead in Sec. 2.4.

ViTs are seen here as separated into feature extraction blocks that use Self-Attention blocks and classifier blocks that rely on Class-Attention blocks (see Fig. 2). A number of modern vision transformers follow this structure like the CaiT (Touvron et al., 2021b) architecture. MixViT is in itself architecture-agnostic and easily generalizes to other ViT architectures with minor adaptations.

### 2.1  MIXVIT OVERVIEW

**At training time**, the $2 \times N$ patches extracted from the 2 inputs are **mixed** into $N$ patches with binary masks $\{\mathcal{M}_i\}$ (so one token per position is picked as shown on Fig. 2) with a ratio $\lambda \sim \beta(\alpha, \alpha)$ (following a corrected CutMix (Yun et al., 2019) scheme here). The $N$ mixed patches are then encoded into $d$-dimensional tokens by a linear layer $e$. Learnable positional embeddings are added to the tokens and the tokens are fed to the $L$ Self-Attention blocks that serve as feature extraction blocks. This yields $N$ attended tokens $t$ but no indication which input/subnetwork each token belongs to.

We propose a novel **source attribution** mechanism $s_i$ to specify which input/subnetwork each patch pertains to. The source attribution adds subnetwork-specific information to the tokens as we elaborate

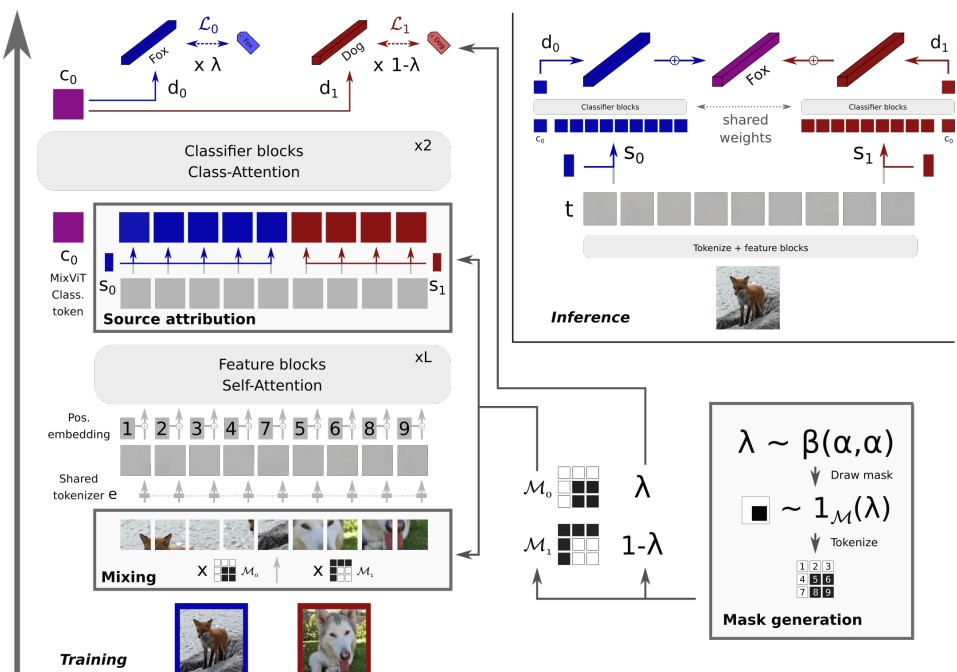

Figure 2: **Overview of our MixViT framework. At training**, we insert a mixing and a source attribution step in the normal flow of the network. Source attribution associates tokens with the relevant input/subnetwork, and can be done through either source encoding or source embedding. **At inference**, we perform multiple passes on the last layer using differently sourced tokens.

on in Sec. 2.3. The sourced tokens $t^*$ are passed - along with a learnable classification token $c_0$ - to the remaining classifier blocks. After the class-attention blocks, we retrieve the classification token's features for classification. 2 predictions $p_i$ are then obtained from this feature representation through the use of $M$ different dense layers $d_i$. We then use the input targets $y_i$ to optimize the subnetworks' cross-entropy losses weighted by the mixing ratio $\lambda$ following standard MIMO practices:

$$\mathcal{L}_{MixViT} = \lambda \cdot \mathcal{L}_0 + (1 - \lambda) \cdot \mathcal{L}_1 = \lambda \cdot l_{CE}(y_0, p_0) + (1 - \lambda) \cdot l_{CE}(y_1, p_1). \qquad (1)$$

**At inference**, MixViT only has to perform feature extraction on the image's patch tokens once as shown on Fig. 2: **no mixing** is needed since the tokens are subnetwork-agnostic. We then collect predictions for each subnetwork $i$ by using $s_i$ to specialize the general feature patch tokens $t$ for the subnetwork through **source attribution**, and then passing these specialized tokens to the classification blocks along with $c_0$ (see Fig. 2). This new "parallel" inference scheme is made possible by MixViT's novel structure and solves issues with MIMO ViTs as discussed in Sec. 2.2.

## 2.2 STRUCTURING MIXVIT TO FACILITATE FEATURE SHARING BETWEEN SUBNETWORKS

We structure MixViT to capitalize on ViTs' propensity to train subnetworks that share features while still retaining the benefits of training with multiple outputs. This restructuration of classical MIMO hinges on our "source attribution" mechanism (see Sec. 2.3), and explicitly separates the model between a core feature extraction block shared between subnetworks and specialized sub-classifiers.

More precisely, we deconstruct the traditional MIMO structure by training a core feature extractor shared by all subnetworks. It takes a single (mixed) input - with no distinction between subnetworks - and outputs a single set of feature tokens: there are no subnetworks in the MIMO sense. Only the last classifier blocks take two source-attributed inputs (see Sec. 2.3) and output two predictions.

This structure both accentuates ViT's natural tendency to share features and enables us to use a new parallel inference scheme. First, as all subnetworks must work from the same pool of feature tokens (after source attribution), they are incentivized to make use of as many useful features as possible.

Moreover, by forcing the core feature extraction blocks to process all patch tokens together - as we give no information on their source - we learn subnetwork agnostic features.

Secondly, this new structure also improves the way the input is processed at inference which is a long standing issue of MIMO frameworks. Indeed, MixViT uses a new "parallel" inference scheme instead of just summing two differently encoded versions of the input: after extracting generic features and attributing sources, we perform one evaluation per set of source attributed tokens "in parallel". This is fairly inexpensive since most computations only have to be performed once (see Sec. 2.4) but would be prohibitively expensive in a standard MIMO framework. Parallel inference is much preferable to the previous "summing" scheme as it forced the use of sub-optimal summing during training and caused significant interference between subnetworks when they share features (Sun et al., 2022). The latter proves particularly problematic in ViTs (shown experimentally in Sec. 4.4).

## 2.3 SOURCE ATTRIBUTION MECHANISMS

We move the subnetworks to the end of vision transformers by introducing source attribution mechanisms $s_i$ after the feature extraction blocks to specialize the **generic patch tokens** $t$ into **subnetwork-specific tokens** $t^*$. As discussed by Rame et al. (2021), distinguishing inputs is necessary for subnetworks to work on their designated input. Standard MIMO frameworks rely on separate input encoders at the beginning of the network (see Fig. 3) to address this. Our source attribution mechanism extends this mechanism for ViTs and can be deployed anywhere in the network.

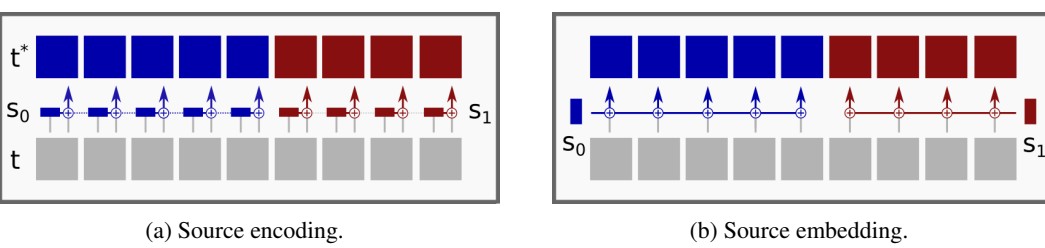

(a) Source encoding.            (b) Source embedding.

Figure 3: Source attribution helps the model identify information relevant to each input.

Source attribution is most naturally implemented with subnetwork-specific **source encoders** (see Fig. 3a). We define our source encoders $s_i$ such that $s_i = Linear(W_i^s; b_i^s)$ are Linear layers. Intuitively, each patch token is projected by the relevant source encoder $s_i$. In practice, we project the feature tokens $t$ with the Linear layers $s_i$ into specialized tokens, and mix these specialized tokens following the masks $\mathcal{M}_i$ used on the inputs. The result is added to the original residual token to obtain our subnetwork-specific token $t^*$ as figured in Eq. 2 ($\odot$ is the Hadamard product):

$$t^* = t + \sum_{i=0}^{1} \mathcal{M}_i \odot s_i(t) = t + \sum_{i=0}^{1} \mathcal{M}_i \odot (t \times W_i^{s\,T} + b_i^s). \tag{2}$$

We also propose an alternative lightweight **source embedding** mechanism (see Fig. 3b) to avoid incurring complications from training additional Linear layers (see Eq. 3). Source embeddings $s_i$ act similarly to positional embeddings, and are $d$-dimensional vectors that are added to feature tokens to specify which subnetwork the patch is relevant to. This therefore only adds $2d$ parameters and $\mathcal{O}(Nd)$ operations which is *negligible* in ViTs. Interestingly, source embeddings are a particular case of source encodings where the weight matrix is null such that $s_i = Linear(0^{d \times d}; b_i^s)$ and

$$t^* = t + \sum_{i=0}^{1} \mathcal{M}_i \odot s_i(t) = t + \sum_{i=0}^{1} \mathcal{M}_i \odot (t \times 0^{d \times d\,T} + b_i^s). \tag{3}$$

## 2.4 REGARDING COMPUTATIONAL OVERHEAD

Deferring the separation of subnetworks to the last layers allows us to perform separate evaluations for each subnetwork at a very low computational cost if we want to train $M$ subnetworks (usually

2). Consider a transformer on a $C$-class problem with $L$ self-attention blocks and 2 class-attention blocks taking $N$ $d$-dimensional patch tokens. **As a rule of thumb**, one MixViT forward roughly incurs $\frac{L+2M}{L+2}\times$ as many computations as a normal model's during inference, and $1\times$ at training.

Indeed, the most intensive computations are tied to Linear $d \times d$ weight multiplications: $d$ is usually much larger than $N$, $C$ or $M$. While source encoding does incur $M$ such computations per token, each Self-Attention layer incurs 6 per token and Class-Attention slightly more than 3 per token: even for our small 7-layer ConViT (see Sec. 4) with $M = 2$ this represents about $\frac{6\times5+3\times2+2}{6\times5+3\times2} \simeq 1.06\times$ multiplications. The amount of additional computations is therefore bounded by the ratio of passes through attention blocks. This approximation could easily be lowered however: it assumes class-attention is as expensive as self-attention which is very pessimistic (see Appendix A for tighter approximations and asymptotic complexity).

## 3   RELATED WORK

**MIMO frameworks as data augmentations**   MIMO CNNs can be seen as an in-manifold mixing samples data augmentation (Rame et al., 2021): they mix two inputs $x_0$ and $x_1$ in the first network manifold with ratio $\lambda$ before (roughly) optimizing a mixed loss $\lambda l_0 + (1 - \lambda)l_1$. MIMO models however remain fundamentally different as they supervise multiple outputs.

Nevertheless, this similarity is even more pronounced in MixViT as mixing is performed directly in patch/pixel space on a corrected mixed input (similarly to the MixToken scheme (Jiang et al., 2021)). MixViT's feature extractor is wholly shared between subnetworks and seems to benefit from this. Furthermore, much of MixViT's improvements on the single input ConViT backbone must be credited to the model having very strong individual subnetworks due to an inherent regularization.

**Mixing in networks**   MixViT builds upon MIMO techniques (Havasi et al., 2021). Beyond simply extending MIMO input encoders, our source attribution also plays a similar role to the "unmixing" strategy proposed by Sun et al. (2022) to allow subnetworks to share features. Contrarily to unmixing however, MixViT does not require extensive tuning and brings significant performance improvements.

Data multiplexing methods (Murahari et al., 2022) use multiple inputs/outputs to accelerate model throughput to allow faster training and inference at a modest cost in performance. While MixViT shares a similar structure as those methods, it is noteworthy that it strongly improves on the base model instead of degrading performance like data multiplexing methods.

Finally, it must be noted that MixViT emulates some ideas from the ensembling literature (Lakshminarayanan et al., 2017; Lee et al., 2015). Indeed, finding ways to train and evaluate multiple models at a discount is a central question of the field (Lee et al., 2015; Wenzel et al., 2020). Common solutions often resolve to share a core network and train a few specialized neural layers at the end of the network (Lee et al., 2015). MixViT closely matches this strategy, with the distinction that we add specialized source attribution mechanisms instead of learning different parameters.

## 4   EXPERIMENTS

We first show our proposed MixViT framework strongly improves on whichever backbone it builds upon in Sec. 4.1 in smaller datasets, before demonstrating in Sec. 4.2 that MixViT significantly improves accuracy on the large scale ImageNet-1k (Deng et al., 2009) dataset. We then investigate feature sharing in MixViT (Sec. 4.3) and its implications. Finally, Sec. 4.4 shows MixViT performs much better than traditional MIMO frameworks on ViTs.

**Setting**   In this paper, we study the CIFAR-10/100 (Krizhevsky et al., 2009), TinyImageNet (Chrabaszcz et al., 2017), ImageNet-100 (Tian et al., 2020), and ImageNet-1k (Deng et al., 2009) datasets. Most experiments use a variant of the ConViT (D'Ascoli et al., 2021) architecture with 5 GPSA blocks and 2 CA blocks with dimension 384 and 12 heads (see Appendix). On the more complex datasets, we also use deeper CaiT (Touvron et al., 2021b) models. Our transformers work on $8 \times 8 = 64$ patches for CIFAR and TinyImageNet, and $14 \times 14 = 196$ patches on ImageNet. We report the best epoch overall ensemble accuracy as $mean \pm std$ over three seeded runs for the best of the two MixViT variants unless specified otherwise.

Table 1: Comparison of MixViT against similarly sized transformer results reported in the literature.

| Models | # Params | CIFAR100 | CIFAR10 | TinyImageNet | ImageNet-100 |
|---|---|---|---|---|---|
| | | Accuracy (%) | Accuracy (%) | Accuracy (%) | Accuracy (%) |
| Reported results (from (.)) | | | | | |
| CCT-7/3x1 (Hassani et al. (2021)) | 4M | 77.1 | 94.8 | - | - |
| DeiT-T Touvron et al. (2021a) (Zhang et al. (2022)) | 5M | 67.5 | 88.4 | - | - |
| PVT-T Wang et al. (2021) (Zhang et al. (2022)) | 13M | 69.6 | 90.5 | - | - |
| ViT Dosovitskiy et al. (2021) (Lee et al. (2021)) | 3M | 73.8 | 93.6 | 57.1 | - |
| PiT-XS Heo et al. (2021) (Lee et al. (2021)) | 7M | 79.0 | 95.9 | 60.3 | - |
| SL-PiT-XS (Lee et al. (2021)) | 9M | 75.0 | 94.2 | 62.9 | - |
| CaiT-XXS-24 Touvron et al. (2021b) (Lee et al. (2021)) | 9M | 76.9 | 94.9 | 64.4 | - |
| SL-CaiT-XXS-24 (Lee et al. (2021)) | 9M | 80.3 | 95.8 | 67.1 | - |
| Dytox (Douillard et al. (2022), joint training) | 10M | 76.0 | - | - | 79.1 |
| Swin-T Liu et al. (2021b) (Liu et al. (2021a)) | 29M | 53.3 | 59.5 | - | 82.7 |
| T2T-ViT-14 Yuan et al. (2021) (Liu et al. (2021a)) | 22M | 65.2 | 84.2 | - | 82.7 |
| CvT-13 Wu et al. (2021) (Liu et al. (2021a)) | 20M | 73.5 | 89.0 | - | 85.6 |
| CvT-13 + $\mathcal{L}_{DrLoc}$ (Liu et al. (2021a)) | 20M | 74.5 | 90.3 | - | 86.1 |
| Our experiments | | | | | |
| MixViT (ours) with ConViT backbone | 12M | $\mathbf{82.4 \pm 0.1}$ | $\mathbf{96.5 \pm 0.2}$ | $\mathbf{70.2 \pm 0.2}$ | $\mathbf{88.8 \pm 0.3}$ |

**Implementation details** We mix inputs using CutMix, and start training subnetworks individually towards the end of training. We largely adapt the DeiT (Touvron et al., 2021a) training settings that are widely used in the literature (Zhang et al., 2022; Touvron et al., 2021b; Lee et al., 2021) (see Appendix). Unless specified otherwise, we therefore train with the AdamW (Loshchilov & Hutter, 2019) optimizer for 150 epochs with weight decay, RandErase (Zhong et al., 2020), AutoAugment Cubuk et al. (2019), stochastic depth (Huang et al., 2016), label smoothing Szegedy et al. (2016) and 3 Batch augmentations (Hoffer et al., 2020). We also apply CutMix Yun et al. (2019) and MixUp (Zhang et al., 2018) but find applying them simultaneously instead of alternatively yields better result. Finally, we keep the 0.001 learning rate but adopt a step decay instead of cosine.

## 4.1 MixViT significantly improves the performance of vision transformers

We start by showing how our MixViT ConViT fares against the current transformer state-of-the-art in Sec. 4.1.1 across 4 datasets, before demonstrating MixViT consistently improves over single-input single-output baselines across multiple backbones in Sec. 4.1.2.

### 4.1.1 MixViT results relatively to similar transformers

We provide here some context on the performance of our models with regard to the existing literature. Tab. 1 gives a comparison to the performance of normally trained transformers of similar sizes. Our results over 150 training epochs should allow a mostly fair comparison: vision transformers are typically trained for comparable or longer amounts of time (100 (Liu et al., 2021a) to 200 Hassani et al. (2021) or more (Zhang et al., 2022) epochs) on CIFAR, TinyImageNet and ImageNet-100.

**CIFAR** Tab. 1 shows MixViT largely outperforms small ViTs. In fact, our MixViT's $83.4\%$ accuracy on the extended 300 epochs setting (see Appendix E) is **the best ViT on CIFAR-100**. Indeed, the closest in the literature are $82.7\%$ for a CCT-7/3x1 (Hassani et al., 2021) trained 5000 epochs and $82.6\%$ for a 90M parameters NesT-B (Zhang et al., 2022) trained 300 epochs.

**TinyImageNet** Tab. 1 demonstrates MixViT largely outperforms transformers on the more complex TinyImageNet dataset. As the best reported transformers on the dataset are in line with our single-input single-output backbones, the gains on the state-of-the-art can largely be attributed to MixViT. Our CaiT-XS MixViT even reaches $70.9\%$ (see Sec. 4.1.2) which **improves on the previous state-of-the-art** held by MixMo's ResNet-18-3 model ($70.2\%$ accuracy), to the best of our knowledge.

**ImageNet-100** Tab. 1 shows MixViT outperforms the competing architectures for comparable training times. This suggests MixViT has little trouble accommodating larger $224 \times 224$ images.

### 4.1.2 MixViT consistently improves models across backbones and datasets

Tab. 2 shows both of our MixViT variants provide significant improvements over their single-input single-output counterpart across a multitude of architectures and datasets. For instance, on

Table 2: MixViT significantly improves the performance of vision transformers

|  | (a) TinyImageNet | |  | (b) ImageNet-100 | |
| --- | --- | --- | --- | --- | --- |
| Backbone | Method | Accuracy (%) | Backbone | Method | Accuracy (%) |
| ConViT | Single-input/output | $65.4 \pm 0.2$ | ConViT | Single-input/output | $86.6 \pm 0.2$ |
|  | MixVit-embedding (ours) | $\mathbf{70.0 \pm 0.1}$ |  | MixVit-embedding (ours) | $\mathbf{88.8 \pm 0.3}$ |
|  | MixVit-encoding (ours) | $\mathbf{70.2 \pm 0.2}$ |  | MixVit-encoding (ours) | $\mathbf{88.6 \pm 0.5}$ |
| CaiT XXS-24 | Single-input/output | $66.2 \pm 0.3$ | CaiT XXS-24 | Single-input/output | $85.8 \pm 0.2$ |
|  | MixVit-embedding (ours) | $68.6 \pm 0.3$ |  | MixVit-embedding (ours) | $\mathbf{87.2 \pm 0.3}$ |
|  | MixVit-encoding (ours) | $\mathbf{69.6 \pm 0.2}$ |  | MixVit-encoding (ours) | $86.7 \pm 0.2$ |
| CaiT XS-24 | Single-input/output | $68.6 \pm 0.1$ | CaiT XS-24 | Single-input/output | $87.6 \pm 0.4$ |
|  | MixVit-embedding (ours) | $69.4 \pm 0.1$ |  | MixVit-embedding (ours) | $\mathbf{89.7 \pm 0.3}$ |
|  | MixVit-encoding (ours) | $\mathbf{70.9 \pm 0.2}$ |  | MixVit-encoding (ours) | $89.3 \pm 0.2$ |

|  | (c) CIFAR-100 | |  | (d) CIFAR-10 | |
| --- | --- | --- | --- | --- | --- |
| Backbone | Method | Accuracy (%) | Backbone | Method | Accuracy (%) |
| ConViT | Single-input/output | $79.5 \pm 0.1$ | ConViT | Single-input/output | $96.1 \pm 0.1$ |
|  | MixViT-embedding (ours) | $\mathbf{82.4 \pm 0.1}$ |  | MixViT-embedding (ours) | $\mathbf{96.5 \pm 0.2}$ |
|  | MixViT-encoding (ours) | $82.1 \pm 0.1$ |  | MixViT-encoding (ours) | $\mathbf{96.5 \pm 0.2}$ |

TinyImageNet, our larger CaiT-XS model goes from a baseline of 68.6% to 70.9% accuracy with MixViT. Interestingly, much of the gains are due to a higher accuracy of the individual subnetworks: the framework also has a strong regularizing effect, possibly due to the feature extractor being shared. This is particularly interesting considering ViTs' notorious difficulties with smaller datasets.

Our lightweight MixViT-embedding outperforms the complete MixViT-encoding on CIFAR (Tabs. 2c and 2d). MixViT-encoding however overtakes it on the more complex TinyImageNet (Tab. 2a). Interestingly, this difference is deepened by both model depth (our ConViT has 7 layers against our CaiTs' 26) and model width (Cait-XXS's 192 against Cait-XS's 288). As such, source embeddings are much faster and easier to train, but source encodings are more powerful overall.

## 4.2 PUSHING THE MODEL ON LARGE-SCALE DATASETS: IMAGENET-1K

Tab. 3 shows MixViT still improves ViTs on the large-scale ImageNet-1k dataset (Deng et al., 2009): simply training 2 subnetworks with MixViT improves a CaiT-M-24's performance by +0.8% which is significant given how challenging improving beyond 82% is on ImageNet-1k. This compares favorably with results reported by traditional MIMO frameworks.

Table 3: ImageNet-1k scores.

| Backbone | Method | Accuracy (%) |
| --- | --- | --- |
| ResNet 50 | Single-input/output | 76.1 |
|  | MIMO (Havasi et al., 2021) | 77.5 |
| ResNet 18-7 | Single-input/output | 77.2 |
|  | MixMo (Rame et al., 2021) | 78.5 |
| CaiT S-24 | Single-input/output | 82.3 |
|  | MixVit (ours) | **82.8** |
| CaiT M-24 | Single-input/output | 82.7 |
|  | MixVit (ours) | **83.5** |

## 4.3 FEATURE SHARING IN MIXVIT

We now verify that subnetworks do share most features in MixViT in Sec. 4.3.1, before showing this leads MixViT to accommodate simpler ViT training settings (see Sec. 4.3.2) and narrow networks (Sec. 4.3.3) contrarily to MIMO CNNs. Additionally, we show in Appendix Sec. B that MixViT is a significant step towards training more than 2 subnetworks at the same time.

### 4.3.1 VISUALIZATION OF CLASSIFIER FEATURES

We check the influence of final features on subnetworks by considering the weight matrices $W_i \in \mathbb{R}^{\#Classes \times \#Feats}$ of our final classifiers $d_i$ (see Fig. 4). A feature's influence is approximated by the L1 norm of $W_i$'s corresponding column: this tallies how much a classifier depends on the feature. We then visualize whether the feature influences both subnetworks or only one on a scatter plot.

Fig. 5 shows MixViT features influence both subnetworks (close to the diagonal). On the contrary, most traditional MIMO features only influence one feature or the other.

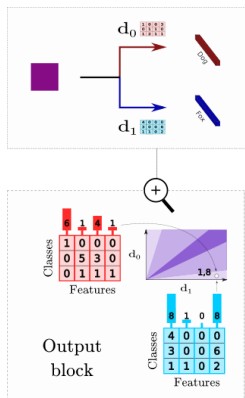

Figure 4: **Feature influence on $d_i$** is given by L1 norm of classifier columns.

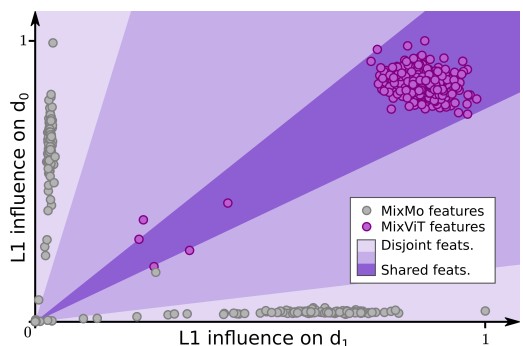

Figure 5: **Visualization of final feature influence on subnetworks** (by proxy on $d_0$ and $d_1$) for MixMo (WideResNet-28-5) and MixViT (ConViT), rescaled to range [0,1]. MixViT features influence both subnetworks (dark purple zone) contrarily to MixMo features.

### 4.3.2 MIXVIT IS LESS RELIANT ON STRONG REGULARIZATION

Typical training schemes for vision transformers often require both heavy augmentation (Yun et al., 2019; Zhang et al., 2018) (in the form of mixing augmentations) and batch augmentations (Cubuk et al., 2019). This is concerning as these procedures can significantly extend training costs and time.

MixViT is not as reliant as traditional ViTs on this strong regularization, possibly because feature sharing in MixViT inherently provides some of the benefits of MSDA and batch augmentation. Indeed, we show mixed inputs to our shared feature extraction blocks which can emulate the effects of MSDA. Moreover, each sample in a batch appears as part of an input pair: it has to be fed to each of the 2 subnetworks once. As such, each of the batch's sample is processed multiple times by the transformer's feature extraction blocks while mixed with other samples.

Tab. 4 shows both MixViT variants prove much more resilient than their single-input single-output backbone on less regularized settings on CIFAR-100. It is particularly interesting to note that even with no batch augmentations and MSDA augmentations, our MixViT-embedding model matches the fully regularized variants of their single-input single-output backbone. This is particularly important as batch augmentation linearly increases computations and has been a key fixture of vision transformer training since its introduction in DeiT (Touvron et al., 2021a). This linear cost has led some modern works to dispense with it (Liu et al., 2021b), but the procedure remains widely used in the literature (Zhang et al., 2022; Wang et al., 2021). As such, MixViT's ability to provide strong performance with no batch augmentation could strongly simplify training schemes.

That is not to say however that MixViT is not compatible with batch augmentation: Fig. 6 shows MixViT still benefits significantly from it. As such, the regularization added by MixViT seems to complement regular batch augmentation well.

Table 4: CIFAR-100 accuracy with MSDA/BA.

| Model | Strong reg. | Accuracy (%) |
|---|---|---|
| Single-input/output ConViT | | $72.3 \pm 0.3$ |
| Single-input/output ConViT | ✓ | $79.5 \pm 0.1$ |
| Loss without strong regularization | | $-7.2$ |
| MixViT (ours) | | $79.0 \pm 0.5$ |
| MixViT (ours) | ✓ | $82.4 \pm 0.1$ |
| Loss without strong regularization | | $-3.4$ |

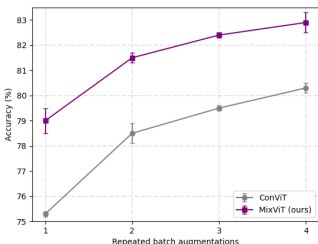

Figure 6: MixViT still scales well with batch augmentation.

### 4.3.3 MixViT adapts to small models better than MIMO CNNs

On the complex TinyImageNet dataset MixViT largely outperforms its much larger CNN-based competition (Tab. 5). Interestingly, this highlights a fundamental difference between traditional MIMO models and our MixViT. MixMo learns subnetworks that share no features (Rame et al., 2021; Sun et al., 2022) and therefore strug-

Table 5: MixViT outperforms CNN-based MixMo models on Tiny-ImageNet.

| Backbone | Method | # Params | Width | Accuracy (%) |
|---|---|---|---|---|
| ResNet 18 | Single-input/output | 11M | 512 | 65.1 |
| | MixMo (Rame et al., 2021) | 11M | 512 | 64.4 |
| ResNet 18-3 | Single-input/output | 100M | 1536 | 69.0 |
| | MixMo (Rame et al., 2021) | 100M | 1536 | 70.2 |
| CaiT XS-24 | Single-input/output | 26M | 288 | $68.6 \pm 0.1$ |
| | MixVit (ours) | 26M | 288 | $\mathbf{70.9 \pm 0.2}$ |

gles to fit good subnetworks on narrow architectures. As such, MixMo needs to consider a wide ResNet-18-3 (He et al., 2016) backbone to start showing improvements. MixViT simply does not have this issue as the subnetworks share features and a large part of the gains come from how the subnetworks learn features they can both use.

## 4.4 Comparison against ViT-based MIMO implementations

MixViT is the first ViT based framework, and was developed to account for the architecture's characteristics. Here, we provide comparisons against applications of previous MIMO methods to ViTs. While these methods were developed for CNN based frameworks, it is possible to transpose the proposed frameworks to Vision Transformers with minimal alterations as detailed in Appendix C.

Tab. 6 shows our direct transpositions of the seminal MIMO and MixMo frameworks fail to perform for ViTs: both frameworks fail to even reach the same performance as the single-input single-output baseline which is consistent with observations in Allingham et al. (2021). MixMo performs much better with our parallel inference scheme, at the cost of significantly more overhead. Even better results are obtained if we only use CutMix mixing in MixMo during training. This strongly suggests issues with sum mixing in MIMO ViTs, possibly due to ViT's propensity to share features.

MixViT strongly improves over the base model with minimal overhead. Even without source attribution and therefore identical subnetworks, it brings strong improvements which clearly shows the benefits of multi-output supervision in a properly designed ViT. Adding source attribution, we recover distinct predictions and ensembling benefits to reach our final 82.4% accuracy.

Table 6: CIFAR-100 accuracy and estimated FLOPS of the backbone and MIMO formulations.

| Backbone | Method | Inference Mixing | # FLOPS | Accuracy (%) |
|---|---|---|---|---|
| | Single-input | N/A | $1\times$ | $79.5 \pm 0.1$ |
| | MIMO | Sum | $1\times$ | $77.1 \pm 0.1$ |
| | MixMo | Sum | $1\times$ | $77.0 \pm 0.2$ |
| ConViT | MixMo w/ parallel inference | Parallel | $2\times$ | $78.5 \pm 0.2$ |
| | Cut-MixMo only w/ parallel inf. | Parallel | $2\times$ | $80.1 \pm 0.2$ |
| | MixViT w/o source attribution (ours) | Parallel | $1.1\times$ | $81.5 \pm 0.2$ |
| | MixViT (ours) | Parallel | $1.1\times$ | $\mathbf{82.4 \pm 0.1}$ |

## 5 Discussion

We propose MixViT, the first MIMO framework for ViTs that leads to significant improvements over the single-input single-output baseline. We take advantage of ViT's innate properties to train subnetworks that share features, thereby solving a longstanding scaling issue in ViTs and adding implicit regularization between subnetworks. We achieve this by restructuring the standard MIMO framework and introducing a novel source attribution mechanism that helps identify input information.

The induced regularization and implicit ensembling obtained by fitting subnetworks leads MixViT to strongly improve on its base model across multiple architectures (ConViT, CaiT) and datasets (CIFAR, TinyImageNet, ImageNet-100, and ImageNet-1k). MixViT models set a new state-of-the-art on TinyImageNet and a new state-of-the-art for transformers on CIFAR-100. Furthermore, the combination of MixViT's feature sharing and multi-output supervision brings strong implicit regularization which can greatly simplify training settings.

**Ethics statement**    At the least, MixViT aims to improve classification systems, to say nothing of possible applications to object detection and other complex tasks. As such, it must be acknowledged that it could lead to significant job automation and nefarious applications. While the implications of job automation can be debated to an extent, it will undeniably lead to a fundamental change in our society. It is also the unfortunate reality that deep learning frameworks can be used by bad actors to harmful ends, with applications like Deepfakes already causing real problems today.

Nevertheless, it is our belief that these neural frameworks can and will have an overall positive impact on the world by easing daily life and improving response rates to critical application. More precisely, we hope MixViT will serve to provide more reliable decision support systems in difficult applications so that human operators can take better and faster critical decisions.

**Reproducibility statement**    We outline all significant components of the MixViT framework in Sec. 2. For contractual reasons it is not possible to give access to the full codebase, although this is still being discussed with the concerned organization at the moment. We have included a detailed summary of experimental settings in Appendix H, and given pseudo-code of the key components as well as indications concerning the codebases we adapted in Appendix I.

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

APPENDIX

This appendix contains the following information:

- In Sec. A, we provide further notes on the overhead incurred by MixViT.
- In Sec. B, we show MixViT suffers much less from training more than 2 subnetworks.
- In Sec. C, we provide detailed descriptions of naive implementations for standard MIMO methods on ViTs used in Sec. 4.4.
- In Sec. D we provide additional ablation experiments on MixViT.
- In Sec. E, we provide additional comparison of MixViT with MixMo on the CIFAR-100 dataset (which is much harder for vision transformers than CNNs).
- In Sec. F, we provide additional baseline comparisons for TinyImageNet and ImageNet-100.
- In Sec. G, we provide information on the licenses of some assets used in this paper.
- In Sec. H, we provide details regarding our experimental protocol.
- In Sec. I, we provide information on our codebase and pseudo-code.

## A    Further notes on MixViT overhead

We provide in Sec. 2.4 a very rough upper bound on the additional overhead incurred by MixViT at inference for one forward evaluation. We provide here a more nuanced discussion on the computations at play and show that - for $M = 2$ or at least bounded - the asymptotic complexity in $L$, $N$ and $d$ of MixViT is the same as that of a normal model.

Similarly to Sec. 2.4, we consider a transformer on a $C$-class problem with $L$ self-attention blocks and 2 class-attention blocks taking $N$ $d$-dimensional patch tokens. We neglect the computational overhead of induced by adding bias terms, embedding the patches into tokens, classifying from the classification tokens features, normalization and source attribution.

**How many computations do class-attention and self-attention need ?**    We assumed previously class-attention and self-attention are as expensive computationally which leads to a very pessimistic upper bound on the overhead.

Let us count the important operations in a Self-Attention layer:

1. We have to extract key, query and value representations for each token which causes $3Nd^2$ operations due to $3N$ weight multiplications.

2. We have to compute the similarity scores between $N$ tokens which means $N^2d$ operations.

3. We must compute the attended representation for all tokens which costs approximately $N^2d$ operations as one attended token requires adding $N$ $d$-dimensional tokens (we take the cost of multiplying the value representation by the attention weight to be atomic, but it is technically $d$).

4. Most implementations include a projection step to properly aggregate the multiple attention heads which means an additional $Nd^2$ cost.

5. Each token is then processed by the same 1 hidden layer multi-layer perceptron. While the hidden dimension of the perceptron can vary it is standard in the literature to take $4d$. Therefore we have to consider $2 \times 4 \times Nd^2$ operations as we consider 2 weights multiplications of dimensions $d \times 4d$.

This yields a cost of approximately $3Nd^2 + N^2d + N^2d + Nd^2 + 8Nd^2 = 13Nd^2 + 2N^2d$ in a Self-Attention layer whereas the cost of a Class-Attention layer is about $3Nd^2 + Nd + Nd + d^2 + 8d^2 = (3N + 9)d^2 + 2Nd$ operations. Indeed, Class-Attention follows a similar outline with the following differences: in 2) we only have $Nd$ operations (similarity to the class token), in 3) we only compute one attended representation for cost $Nd$, in 4) we only compute the projection for one token at cost $d^2$ and in step 5) we only apply the MLP to one token for $8d^2$ operations.

If we throw out the terms linear in $d$ (as it is usually the largest term by far), Class-Attention therefore costs $\frac{3N+9}{13N}\times$ as many operations as Self-Attention layer. Note that - to the best of our knowledge - $N = 64$ at the least in the literature, so this ratio is closer at a maximum $0.25\times$.

**Tighter approximation of MixViT's inference overhead**    We can now revise the very pessimistic approximation given Sec. 2.4 provided the estimations that Class-Attention costs at most $0.25\times$ as many computations as Self-Attention. One forward evaluation of MixViT costs approximately $\frac{L+0.5M}{L+0.5}\times$ as many operations with this. For our modified ConViT, this indicates a ratio of only $1.1\times$.

**Asymptotic complexity in $L$, $N$ and $d$**    Asymptotically, for a fixed number of subnetworks (M=2 usually), the complexity of MixViT and the underlying transformers are identical and equal to $\mathcal{O}(L(Nd^2 + N^2d))$. Indeed, as can be seen from the previous calculations, Self-Attention has a complexity of $\mathcal{O}(N^2d + Nd^2)$ and Class-Attention $\mathcal{O}(Nd^2)$. A normal forward pass is therefore on the order $\mathcal{O}(L(Nd^2 + N^2d))$ as we have $L + 2$ attention layers. MixViT-encoding induces $M$ computations on the same order in the form $O(Nd^2)$ source encodings. MixViT-encoding's complexity is therefore $\mathcal{O}((L+M)Nd^2 + LN^2d)$ or $\mathcal{O}(L(Nd^2 + N^2d))$ if we consider $M$ bounded.

## B    MIXVIT ACCOMMODATES MULTIPLE SUBNETWORKS MUCH BETTER

The M>2 subnetworks case has been an open problem ever since the seminal MIMO (Havasi et al., 2021) paper. We believe MixViT is a significant step towards solving this problem, which has been a pain point for all existing MIMO techniques (Havasi et al., 2021; Rame et al., 2021; Sun et al., 2022). For instance, with a single input baseline around 80% accuracy, a WideResNet-28-5 MixMo network shows a strong deterioration of performance over CIFAR100 (batch repetition 2) whereas a ConViT MixViT does not as shown on Fig. 7.

It seems the feature sharing in MixViT likely helps the network train more than 2 independent subnetworks since our performance for M=4 is about 9% better with MixViT (over MixMo), and remains at a steady 3% above the single-input baseline. While this is beyond the scope of this paper, we believe a properly designed MSDA scheme could allow MixViT to see further improvements with M>2 subnetworks.

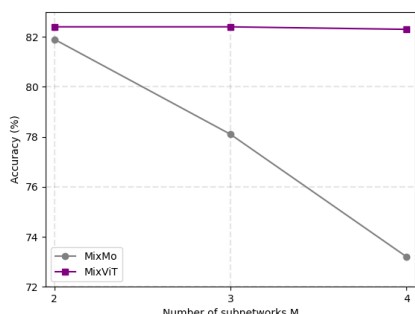

Figure 7: MixViT deals much better with more than 2 subnetworks.

As discussed in MixMo (Rame et al., 2021), MIMO frameworks can generalize to more than 2 inputs by simply using a MSDA method that mixes more than 2 inputs. The training objective in this case is simply the sum of the subnetwork losses ponderated by the (reweighted) mixing ratios. While such methods have been attempted in the literature, they work fairly poorly for MIMO techniques.

A simple MSDA for this open problem could be to pick at random 2 inputs and perform cut-mixing on those two inputs (the mixing ratio for the other inputs would be 0). This is the scheme we use for the previous results on MixMo and MixViT.

# C   NAIVE IMPLEMENTATION OF MIMO AND MIXMO TRANSFORMERS

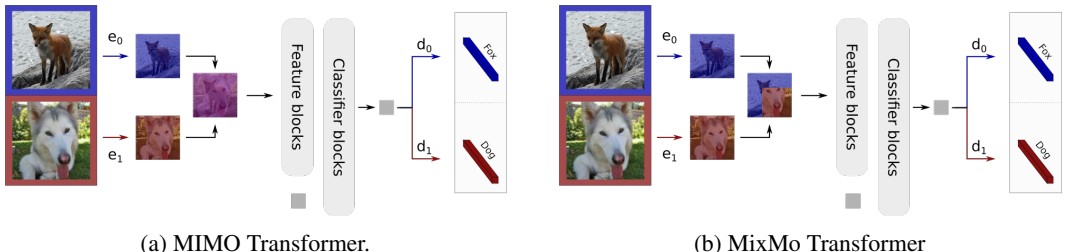

(a) MIMO Transformer.                                        (b) MixMo Transformer

Figure 8: Transposition of traditional MIMO architectures to vision transformers.

At first blush, MIMO architectures can easily be transposed to vision transformers by analogy with convolutional neural networks. We outline below how vision transformers can replicate the steps of MIMO models, and provide an illustration of MIMO transformers in Fig. 8.

We consider $M$ (inputs, labels) pairs $\{(x_i, y_i)\}_{0 \le i < M}$ during training: **we feed $M$ inputs to the model as a combined input**. To this end, we break down each image into small $4 \times 4$ patches and embed the patches into $N$ 384-dimensional tokens using $M$ linear embedding layers $e_i$ (one for each input). Corresponding tokens (at the same position in the two images) are then mixed into one common set of tokens (either through sums Havasi et al. (2021) or other mixing schemes Rame et al. (2021)). The core network (feature blocks + classification blocks) then processes this representation until a final attended classification token is obtained. $M$ dense layers $\{d_i\}_{0 \le i < M}$ - one for each subnetwork - then yield $M$ predictions from this attended classification. At test time, $M$ predictions can be obtained for one image by embedding the image patches with the $M$ linear embeddings and summing the sets of embedded patch tokens.

In the seminal MIMO framework, we mix the embedded tokens by simply summing them. The CutMix-based MixMo formulation interpolates between the tokens following either a randomly drawn square mask with ratio $\lambda \sim \beta(\alpha, \alpha)$, or a linear scheme (similar to summing).

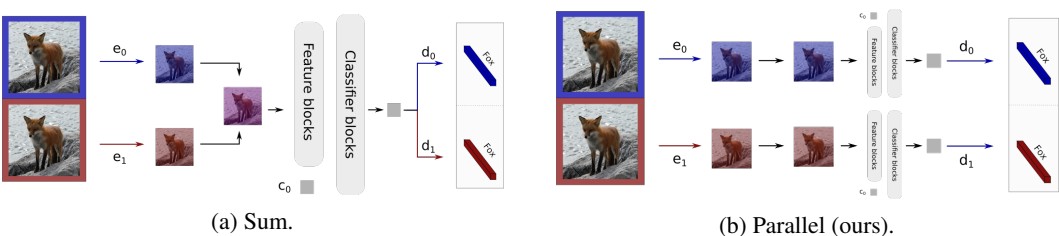

(a) Sum.                                                     (b) Parallel (ours).

Figure 9: Inference schemes for MIMO architectures.

# D  FURTHER ABLATIONS

We verify in Tab. 7 the benefits of our source attribution design (Sec. D.1), our inference scheme (Sec. D.2) and our choice to mix inputs (Sec. D.3) on CIFAR-100.

## D.1  EFFECT OF SOURCE ATTRIBUTION

Tab. 7 shows MIMO training seems to inherently benefit the model: networks trained without source attribution are significantly better than standard ConViT networks. While such models learn completely identical subnetworks (as there is no source attribution), having to learn features useful for the classification of multiple images simultaneously provides ample regularization.

We separate subnetworks in the last layers instead of separately encoding the inputs early in the network to avoid having to perform multiple forward passes for inference (see Sec. 4.4). We however find in Tab. 7 that displacing the separation back to early in the model slightly deteriorates performance: the model benefits from wholly sharing feature blocks between subnetworks.

Table 7: CIFAR-100 accuracy and estimated FLOPS for ablations on components and implementation choices in the MixViT framework. Default settings are highlighted in grey for reference.

| Method | # FLOPS | Accuracy (%) |
|---|---|---|
| Importance of source attribution (Sec. D.1) | | |
| Late source embedding | ×1.1 | 82.4 ± 0.1 |
| Late source encoding | ×1.1 | 82.1 ± 0.1 |
| No source attribution | ×1.1 | 81.5 ± 0.2 |
| Early input source embedding | ×2 | 82.2 ± 0.2 |
| Early source encoding | ×2 | 82.2 ± 0.2 |
| Inference scheme (Sec. D.2) | | |
| Separate inference | ×1.1 | 82.4 ± 0.1 |
| Sum inference | ×1 | 81.9 ± 0.1 |
| Alternating inference | ×1 | 82.0 ± 0.4 |
| Single subnetwork inference | ×1 | 82.0 ± 0.2 |
| Mixing scheme (Sec. D.3) | | |
| Binarized CutMix | ×1.1 | 82.4 ± 0.1 |
| All tokens | ×2 | 81.5 ± 0.2 |

## D.2  INFERENCE SCHEME

In stark contrast to our findings in Sec. 4.4, we find summing the 2 source attributed versions of each patch token at the end of our network before performing a single forward pass yields reasonably good results (Tab. 7). While these results are indeed worse than separate inferences, they are also a far cry from the disastrous MixMo ConViT results in Sec. 4.4. We posit late patch representations are therefore significantly more suited than early patch representations to this sort of combination. Interestingly, we observe similar performance when drawing at random which patches should be attributed to which subnetwork. It is also worth noting that the regularization induced by our MixViT scheme is strong enough for the performance of individual subnetworks to be reasonable on its own.

## D.3  INPUT MIXING SCHEME

We initially chose to compress the inputs into a single set of $N$ tokens through a CutMix scheme to avoid the quadratic cost that would come with using all $M \times N$ tokens from the $M$ inputs. Tab. 7 actually indicates that far from deteriorating performance, our compressed scheme performs better than a model trained with all tokens. We explain this by the fact the CutMix compression introduces additional regularization on the shared feature extractor and slightly more diverse subnetworks.

# E    FURTHER COMPARISON TO MIXMO ON CIFAR-100

Tab. 5 shows MixViT matches the similarly sized WideResNet-28-5 Zagoruyko & Komodakis (2016) on CIFAR-100 when given the same training budget. While MixMo's Rame et al. (2021) reported WideResNet-28-5 performance is better than our default setting MixViT performance, MixMo models train over 300 epochs with batch repetition 4. Batch repetition functions very similarly to Batch augmentation and is often needed to obtain good results with CNN-based MIMO models (which is less so the case for MixViT as per Sec. 4.3.2). If we reduce the learning rate and train with 4 batch augmentations, we close the gap in performance. This

Table 8: Comparison between MixMo and MixViT on CIFAR-100. * indicates 300 training epochs with 4 batch augmentations.

| Models | # Params | Accuracy (%) |
|---|---|---|
| WideResNet-28-5* | 9M | 82.6 |
| WideResNet-28-5 MixMo* | 9M | **83.3** |
| ConViT D'Ascoli et al. (2021) | 12M | $79.5 \pm 0.1$ |
| MixViT w/ ConViT | 12M | $82.4 \pm 0.1$ |
| ConViT* D'Ascoli et al. (2021) | 12M | $81.4 \pm 0.2$ |
| MixViT w/ ConViT* | 12M | $\mathbf{83.2 \pm 0.2}$ |

is particularly noteworthy as vision transformers are typically disadvantaged against ConvNets on the CIFAR datasets as can be seen by the fact our baseline single-input single-output ConViT model only reaches $81.1\%$ accuracy (vs. $82.6\%$) even on this longer training schedule.

# F FURTHER COMPARISONS ON TINYIMAGENET AND IMAGENET-100

We provide additional points of comparison for TinyImageNet and ImageNet-100 as Tab. 1 is fairly limited for these two datasets.

Table 9: Comparison of MixViT against other transformers on TinyImageNet.

| Models | # Params | TinyImageNet |
| --- | --- | --- |
| | | Accuracy (%) |
| Reported results (from [.]) | | |
| ViT Dosovitskiy et al. (2021) (Lee et al. (2021)) | 3M | 57.1 |
| T2T Yuan et al. (2021) (Lee et al. (2021)) | 7M | 60.6 |
| PiT Wang et al. (2021) (Lee et al. (2021)) | 7M | 60.3 |
| Swin Liu et al. (2021b) (Lee et al. (2021)) | 7M | 60.9 |
| CaiT-XXS-24 Touvron et al. (2021b) (Lee et al. (2021)) | 9M | 64.4 |
| SL-CaiT-XXS-24 (Lee et al. (2021)) | 9M | 67.1 |
| Our experiments | | |
| MixViT | 12M | **70.2 ± 0.2** |

Table 10: Comparison of MixViT against other transformers on ImageNet-100.

| Models | # Params | ImageNet-100 |
| --- | --- | --- |
| | | Accuracy (%) |
| Reported results (from [.]) | | |
| Dytox (Douillard et al. (2022), joint training) | 10M | 79.1 |
| T2T-ViT-14 Yuan et al. (2021) (Liu et al. (2021a)) | 22M | 82.7 |
| T2T-ViT-14 + $\mathcal{L}_{DrLoc}$ (Liu et al. (2021a)) | 22M | 83.7 |
| Swin-T Liu et al. (2021b) (Liu et al. (2021a)) | 29M | 82.7 |
| Swin-T + $\mathcal{L}_{DrLoc}$ (Liu et al. (2021a)) | 29M | 84.0 |
| CvT-13 Wu et al. (2021) (Liu et al. (2021a)) | 20M | 85.6 |
| CvT-13 + $\mathcal{L}_{DrLoc}$ (Liu et al. (2021a)) | 20M | 86.1 |
| Our experiments | | |
| MixViT | 26M | **89.7 ± 0.3** |

## G   INVENTORY OF LICENSED ASSETS USED IN THIS WORK

We built upon a number of assets in this work, many of which are under open-source license. We provide below an inventory of the most important licensed assets we used:

- We used the Python programming language under the PSF License agreement. We additionally made use of many python libraries, the most important of which is the pytorch library under BSD-3 Clause license.

- We used a number of models and codebases, most notably the DeiT, ConViT and timm codebases under Apache 2.0 license.

- We used multiple datasets, among which are the CIFAR datasets under a MIT License and a subset of the ImageNet dataset under a BSD-3 Clause license.

- We used the strong data augmentation protocol AutoAugment under the MIT License.

## H   EXPERIMENTAL SETTING

We give here a detailed inventory of hyperparameters and training procedure used in this paper. For contractual reasons, it is not possible to provide the codebase used in this paper, but Appendix I contains pseudo-code and links to the public codebases we built upon.

All our experiments were run three times on three fixed seeds from the same codebase. Quantitative results are given in the form of $mean \pm std$ over the three runs.

**FLOPS estimations**   We provide an estimate of the relative number of FLOPS used by different methods in Tab. 6 and Tab. 7. This number is traditionally computed for pytorch models on a theoretical basis by standardized libraries like `ptflops` or `fvcore`. These libraries however do not support custom attention layers and are therefore un-usable for our purposes.

Fortunately, estimating the ratio of FLOPS is straightforward in Tab. 6: the additional overhead of MIMO and MixMo is known to be approximately $1\times$ on CNNs (which also holds on transformers) and separate inference just leads to inferring twice, hence the $2\times$ ratio. We use the upper bound derived in Appendix A for MixViT and its variants that follow similar computational flows. For the last line of Tab. 7, we give a rough approximation derived by considering the impact of the larger number of tokens on the attention layers.

**Corrected CutMix Scheme**   An issue that arises when considering a CutMix scheme to mix patch tokens is that CutMix masks have binary values for each pixel value whereas a patch contains multiple patches. It stands to reason therefore that we will often have situations where the patch contains both masked pixels and unmasked pixels. While the natural solution is simply to take a continuous mask value by averaging over the patch, we have seen in Sec. 4.4 that subnetworks have trouble sharing tokens.

To address this, we correct the CutMix scheme by using a majority vote within the patch: if most pixels are masked, then we mask the patch and vice-versa. Ties (where we have as many masked and unmasked pixels) are broken by choosing randomly for one set of inputs which input will be assigned the problematic patches. It worth noting that this scheme is for all intents and purposes the MixToken Jiang et al. (2021) scheme, with some practical differences in the sort of mask generated.

**Architectural details**   We used a modified variant of the ConViT D'Ascoli et al. (2021) architecture for our experiments hidden dimension 384, 12 attention heads, 5 GPSA blocks and 2 SA Blocks similarly to dytox Douillard et al. (2022). Patch size was taken to be $4 \times 4$ on CIFAR, $8 \times 8$ on TinyImageNet and $16 \times 16$ on ImageNet-100. It is worth noting GPSA blocks should in theory have a square number of heads to exactly emulate convolutional kernels at initialization, but we did not find this to be an issue experimentally.

We took the CaiT models directly from the seminal paper Touvron et al. (2021b) and used the same patch size as for our ConViT model (which also coincides with CaiT's default settings).

**Training details**  In general, we train our models over 150 epochs using the AdamW optimizer with a learning rate of 0.001, weight decay of 0.05 and linear warmup over 10% of training for most datasets. We follow a step decay schedule with decay rate 0.1 at steps $\{125, 140\}$ For the ImageNet dataset, we only trained for 133 epochs (CaiT-S-24, decay at steps $\{75, 90\}$) and 100 epochs (CaiT-M-24, decay at steps $\{100, 120\}$) due to logistic constraints.

We used a base batch size $B$ of 128 on CIFAR and TinyImageNet and 1048 on ImageNet. The effective batch size $b$ used was dependant on memory constraints, and the learning rate was scaled accordingly with a ratio $\sqrt{\frac{b}{B}}$ following heuristics derived from Granziol et al. (2020).

Following DeiT, we train with a lot of regularization. We use 0.05 weight decay, 0.05 stochastic depth for CaiT models, 0.1 label smoothing, 3 Batch augmentations. We use the CIFAR configuration of AutoAugment for CIFAR and TinyImageNet, and the ImageNet configuration for ImageNet-100. We also apply CutMix Yun et al. (2019) and MixUp Zhang et al. (2018) but find applying them simultaneously instead of alternatively yields better result and is more stable. Practically speaking, we draw CutMix mask with $\alpha = 1.0$ but instead of the mask having binary values (0 or 1), it has continuous values like a $\alpha = 0.8$ MixUp scheme.

Similarly to MixMo, we progressively adapt the input mixing scheme to reflect the mechanism used at inference. In the MixMo case, this means using a summing scheme instead of CutMix with a probability $p$ that linearly grows over the last twelfth of training. In the MixViT case, we use masks that take all tokens from one of the inputs with a probability $p$ that linearly grows over the last twelfth of training. We directly inherited the mechanism from MixMo and did not tune the point at which we start increasing the probability $p$ linearly.

**Choice of hyperparameters**  We tuned the parameter $\alpha \in \{0.25, 0.5, 1.0, 2.0\}$ used to draw MixViT's input mixing CutMix mask on CIFAR-100 and kept it fixed for all other experiments. $\alpha = 0.5$ was chosen as it was more stable and not necessarily for optimality purposes. We did not tune the root parameter used for loss balancing in MixMo and instead discarded it early on in our preliminary experiments (equivalent to choosing root $r = 1$).

While cosine decay is standard in the transformer literature, we failed to converge with it on CIFAR-100 preliminary experiments. As we wished to ensure our single-input single-output backbone had converged for fair comparison with MixViT, we adopted a step decay schedule and roughly set decay steps by considering the training loss on CIFAR-100. Similarly, we train on 150 epochs instead of 100 because our models clearly failed to converge in 100 epochs on CIFAR-100.

We largely inherited all other parameters(patch sizes, batch sizes, 0.001 learning rate, 3 batch augmentations, 0.05 weight decay, AdamW optimizer, 0.05 stochastic depth dropout on CaiT, 0.1 label smoothing, CutMix $\alpha = 1.0$, MixUp $\alpha = 0.8$, RandErase and AutoAugment configurations) from the literature Touvron et al. (2021a); Lee et al. (2021). We did double weight decay and stochastic depth on TinyImageNet and ImageNet-100 for MixViT to avoid overfitting and checked the performance of the backbone with those parameters, but did not look for an optimal value.

## I  PSEUDO-CODE AND CODEBASES

Our experiments were run through a codebase implemented using the PyTorch Paszke et al. (2019) library. Our codebase built upon the public official implementation of MixMo (`https://github.com/alexrame/mixmo-pytorch`) and adapted components from multiple public codebases:

- We took ConViT building blocks from `https://github.com/facebookresearch/convit/blob/main/convit.py`

- We took CaiT building blocks from `https://github.com/facebookresearch/deit/blob/main/cait_models.py`

- We took our AutoAugment implementation from `https://github.com/DeepVoltaire/AutoAugment`

- We took a number of primitive functions from the timm library `https://github.com/rwightman/pytorch-image-models`

The MixMo codebase in particular is well documented and should provide everything necessary to train multi-input multi-output networks. The main adaptation that must be made is the addition of the transformer networks and specific MixViT routines in the transformer networks.

To facilitate this last point, we provide in Alg. 1 and Alg. 2 the training and inference procedure for MixViT. `PatchEmbed` simply embeds the inputs into patch representations as is standard in transformer frameworks, `MixWithMasks` simply mixes the inputs following the masks given, `AddPosEmbed` just adds the positional embeddings to the mixed tokens. `SourceAttribution` implements Eq. 2 or Eq. 3 depending on which variant of MixViT is considered. `SourceAttributionTiled` is similar, except it creates $M$ sets of tokens, each equivalent to `SourceAttribution`$(x, \mathcal{T}_i)$ with $\mathcal{T}_i$ being made up null masks $\mathbf{0}_N$ except for the one at position $i$ being a unit mask $\mathbf{1}_N$.

---

**Algorithm 1:** MixViT forward at training.

**Input:** Inputs $x_i$, Masks $\mathcal{M}_i$
**Parameters:**
Self-Attention blocks $SA_k$
Class-Attention blocks $CA_k$
Classification token $c_0$
Dense classification layers $d_i$
**Result:** Network predictions $p_i$
**Code:**

1 # Format the input;
2 $\{x_i\} := \{\texttt{PatchEmbed}(x_i)\}$;
3 $x := \texttt{MixWithMasks}(\{x_i\}, \{\mathcal{M}_i\})$;
4 $x := \texttt{AddPosEmbed}(x_i)$;
5 # Standard forward evaluations;
6 **for** $k=0,...,L\text{-}1$ **do**
7 $\quad x := SA_k(x)$;

8 # Associate patches and subnetwork/input;
9 $x := \texttt{SourceAttribution}(x, \{\mathcal{M}_i\})$;
10 # Standard class-attention;
11 $x := \text{Concat}([c_0;x])$;
12 **for** $k=0,1$ **do**
13 $\quad x := CA_k(x)$;
14 # Predict from features;
15 $c_0 := x[0]$;
16 $\{p_i\} := d_i(c_0)$;
17 Return $\{p_i\}$

---

**Algorithm 2:** MixViT forward at inference.

**Input:** Inputs $x_i$
**Parameters:**
Self-Attention blocks $SA_k$
Class-Attention blocks $CA_k$
Classification token $c_0$
Dense classification layers $d_i$
**Result:** Network predictions $p_i$
**Code:**

1 # Format the input;
2 $\{x_i\} := \{\texttt{PatchEmbed}(x_i)\}$;
3 # No mixing needed !;
4 $x := \texttt{AddPosEmbed}(x_i)$;
5 # Standard forward evaluations;
6 **for** $k=0,...,L\text{-}1$ **do**
7 $\quad x := SA_k(x)$;

8 # Create sets of sourced tokens;
9 $\{x_i\} := \texttt{SourceAttributionTiled}(x)$;
10 # Standard class-attention;
11 $\{x_i\} := \{\text{Concat}([c_0;x_i])\}$;
12 **for** $k=0,1$ **do**
13 $\quad \{x_i\} := \{CA_k(x_i)\}$;
14 # Predict from features;
15 $\{c_{0,i}\} := \{x_i[0]\}$;
16 $\{p_i\} := \{d_i(c_{0,i})\}$;
17 Return $\{p_i\}$

---

