# OpenReview forum: "Reconciling feature sharing and multiple predictions with   MIMO Vision Transformers"
_ICLR.cc/2023/Conference — Submitted to ICLR 2023_

### Official Review · Reviewer_sVL6 · 2022-10-15

**Confidence:** 4
**Correctness:** 2
**Technical Novelty And Significance:** 2
**Empirical Novelty And Significance:** 2
**Recommendation:** 5

**Clarity, Quality, Novelty And Reproducibility:**

The proposed method is not clearly explained, and the figures do not really help. Most captions are only a sentence or so – see e.g. Figure 3. This makes the figures hard to interpret. I would encourage the authors to add some PyTorch/NumPy/Jax pseudo-code for the network architecture.

For table 1, the hyperparameters might be so different that it is hard to compare the methods. The number of parameters can also be misleading since many authors change the resolution when training networks – and higher resolution typically leads to better scores.

I would also like to ask, are all numbers in table 3 from you? Do the methods in table 3 all use the same hyperparameters? If not, the different performances might simply come from different training recipes instead of the MIMO architecture. I would encourage the authors to use the original code from the VIT authors and then make sure that the proposed method can bring clear improvements over the official code. I am not convinced that the proposed method would actually improve SOTA methods. The best result would be to improve the SOTA number for e.g. Vit-base on imagenet.

Mixing two input images on the token level seems like a simple adaptation of cut mix. Could you add some ablation experiments that test how the method compares against cut mix and if the improvements from the two methods are orthogonal?



**Strength And Weaknesses:**

Strengths:

* The proposed method generally gives solid gains in known benchmarks.
* The authors give clear ablation experiments which show that their methods are better than other MIMO methods for VITs (table 6).

Weaknesses:

* The novelty is low.
* Some results do not control for the training recipe, e.g. numbers in table 1 might have vastly different hyperparameters.
* It is not clear if the method improves SOTA models.
* The paper mostly focuses on small-scale datasets. It is known that VITs work best for large-scale datasets.
* The clarity when explaining the proposed architecture could be improved.


**Summary Of The Paper:**

The paper considers the use of the multiple-input multiple-output (MIMO) paradigm for vision transformers. The authors propose to train a ViT by using two images A and B at a time – the two images are tokenized and the tokens are “mixed” by randomly selecting tokens from image A or B. The mixed tokens are then passed through the ViT, and a proposed “source attribution” layer at the top allows the network to predict the classes of both images A and B. Experimentally, the authors show that 1) their network beats previous small networks reported in the literature on small-scale datasets (table 1), 2) the proposed MIMO architecture gives improvements across datasets and architectures (table 2 and 3) and 3) that the proposed MIMO architecture beats previous MIMO architectures for ViTs (table 6).



**Summary Of The Review:**

The authors present a natural extension of the MIMO framework for ViTs and show how this can improve performance for various vision datasets. While the empirical results are positive, it is not clear that all factors are controlled for and that the proposed method would actually improve SOTA models. Furthermore, the novelty is relatively low and the proposed method is not explained clearly.

---

> ### Author Response · Authors · 2022-11-10
> **Response to reviewer sVL6 (round 1) 1/2**
>
> Thank you for your feedback and comments.
>
> As noted in our general comments, we are very concerned by a **fundamental misunderstanding** in the **motivation** of this paper. **This paper sets out to address a core issue in the existing MIMO frameworks (lack of feature sharing between subnetworks)** as stated in the title, introduction and throughout the paper. MixViT achieves this by considering a new framework on ViTs and proposing a number of adaptations. Incidentally, experimental results (Sec. 4.4) show our MixViT is the only working MIMO framework on ViTs. As such, the primary goal of the paper **is absolutely not** to adapt MIMO models to ViTs. Rather, it is a step we take towards our goal of feature mutualization in MIMO models.
>
> It is therefore surprising to us that your review makes no mention of feature sharing/mutualization at all.
>
> Regarding your comments:
>
> **1) The novelty is low.**
>
> This is of course left to your own appreciation. We would however like to point out the paper sets out to solve the issue of feature mutualization in MIMO models. It achieves this by studying MIMO ViTs, completely changing the traditional MIMO structure, and introducing source attribution. In particular, the new structure introduces a new parallel inference scheme only possible on this new formulation that appears necessary for MIMO frameworks to work on ViTs (Tab. 6).
>
> **2) Some results do not control for the training recipe, e.g. numbers in table 1 might have vastly different hyperparameters.**
>
> **For table 1, the hyperparameters might be so different that it is hard to compare the methods. The number of parameters can also be misleading since many authors change the resolution when training networks – and higher resolution typically leads to better scores.**
>
> This is a fair point. We did strive to show settings with equivalent or better conditions, but the settings are indeed different so these results should mostly be considered for context as we already mention in Sec. 4.1.1. We would however  like to point out showing such results for reference is fairly common practice in the field (e.g. Douillard et al. (2022)).
>
> There is one point of reference we think is particularly important for the comparison. The best CaiT-XXS-24 (SL setting) reported in the literature on TinyImageNet has 67.1% accuracy vs. 66.2% for our own CaiT-XXS-24 baseline. MixViT (with CaiT-XXS-24) very substantially improves on this score with an accuracy of 69.6% in what we believe is a mostly fair comparison.
>
> We personally use the lowest image resolution/grid size used on the datasets we consider as discussed in our experimental details: we have no advantage on this front. If anything, we actually suffer from the fact larger resolution can improve results (as we do not take advantage of this trick).
>
> **3) It is not clear if the method improves SOTA models.**
>
> On most backbone/dataset tested (Tab. 2 and 3), we improve upon well-established single-input/output baselines by an order of magnitude larger than the standard deviation. The architectures we consider (CaiT and a ConViT variant) are well-established ViT architectures.
>
> When comparing single-input/output baselines to MixViT variants (Tab. 2 and 3), all experiments are run on the same settings (or at least with hyperparameters picked from the same pool).
>
> **4) The paper mostly focuses on small-scale datasets. It is known that VITs work best for large-scale datasets.**
>
> We do provide results for ImageNet which is the gold standard for large-scale classification dataset.
>
> Additionally, we would like to point out small datasets are a typical pain point for Vision Transformer and have been the target of specialized study (Lee et al., 2021; Liu et al., 2021a). As such, we feel it is noteworthy that MixViT helps train particularly strong Vision Transformers on such small datasets.
>
> **5) The clarity when explaining the proposed architecture could be improved.**
>
> **The proposed method is not clearly explained, and the figures do not really help. Most captions are only a sentence or so – see e.g. Figure 3. This makes the figures hard to interpret. I would encourage the authors to add some PyTorch/NumPy/Jax pseudo-code for the network architecture.**
>
> That is very possible, and we would be more than happy to improve the writing on this part. In particular, we will include larger captions in future revisions. Which parts do you feel are particularly problematic?
>
> We would like to point out pseudo-code is provided in Appendix I.

---

> > ### Author Response · Authors · 2022-11-10
> > **Response to reviewer sVL6 (round 1) 2/2**
> >
> > **6) I would also like to ask, are all numbers in table 3 from you? Do the methods in table 3 all use the same hyperparameters? If not, the different performances might simply come from different training recipes instead of the MIMO architecture. I would encourage the authors to use the original code from the VIT authors and then make sure that the proposed method can bring clear improvements over the official code. I am not convinced that the proposed method would actually improve SOTA methods. The best result would be to improve the SOTA number for e.g. Vit-base on imagenet.**
> >
> > All the ViT results are. We report the official MIMO and MixMO (CNN based) results for context. While you could argue the settings are different, the bigger disadvantage for CNNs is really that ViTs are better than CNNs on ImageNet to begin with. It is the relative gain between baseline and MIMO models we aim to highlight with this comparison.
> >
> > We wholeheartedly agree fair comparison against the baseline requires using the same settings, which is why we were careful to use the same settings (up to some hyperparameters tuned on the same pool) between the baselines and MixViT in Tab. 2 and 3 as discussed in our answer to 3).
> >
> > On the issue of code, all architecture code was directly imported from official repositories and implementations as discussed in Appendix I.
> >
> > **7) Mixing two input images on the token level seems like a simple adaptation of cut mix. Could you add some ablation experiments that test how the method compares against cut mix and if the improvements from the two methods are orthogonal?**
> >
> > We do not particularly claim our tokenized CutMix is better than standard CutMix to mix inputs in MIMO models. It is simply done to accommodate our source attribution to Cut-MixMo.
> >
> > If the question is about using CutMix to mix inputs at the pixel level vs. our mixed tokens in MixViT, training a ViT with pixel level mixing leads to accuracy 79.5% vs. 81.5% for MixViT without pixel level mixing (all other things equal). The former uses CutMix on the input, whereas the latter uses Tokenized CutMix  at the token level.
> >
> > Of course, the two cumulate quite well since the full MixViT model (with pixel level mixing on the inputs) has an accuracy of 82.4%.

---

### Official Review · Reviewer_jCdb · 2022-10-25

**Confidence:** 5
**Correctness:** 3
**Technical Novelty And Significance:** 2
**Empirical Novelty And Significance:** 2
**Recommendation:** 3

**Clarity, Quality, Novelty And Reproducibility:**

The clarity is good, novelty is limited as the method is a direct adaption of existing MIMO to vision transformers. Reproducibility is good as authors provides many implementation details.

**Strength And Weaknesses:**

 Strength:
		Many small-scale datasets are used for validating the results. And the paper proposes one working example of MIMO for vision transformers.

	Weaknesses:
		The paper introduces one way of bring MIMO to vision transformers, which is quite straightforward and novelty-limited,
Source attribution is a direct analogy of using different input projection in existing MIMO.  But there exist many other natural designs that can potentially work for vision transformers. The study space in the paper is not enough to show this proposed way is better.  i.e. important baselines are missing. For example, use different patch embedding for different inputs (similar to different project for inputs in MixMo paper).
		Again, this specific design of late source embedding is not thoroughly studied. Early source embedding with carefully designs comparison is missing. Or middle source embedding, etc, this design should be carefully studied.
		Experiments are mainly conducted on small-scale datasets, ability to adapt to other large-scale datasets and large models remain unknown. Table 3 shows CaiT S-24 and M-24 results are 82.3 and 82.7. However, table 3 in CaiT paper shows s-24, m-24 results are 82.7 and 83.4 respectively. Can authors check the numbers here? And provides some large model (CaIT-M, or other large popular ViT variants) results would be better.


**Summary Of The Paper:**

The paper proposes to bring MIMO for CNNs to vision transformers. Specifically, it designs a source attributing module at last layer to separate the input source and perform classification for different input. Experiments are mainly conducted on small-scale datasets, CIFAR, Imagenet-100 to show its effectiveness.


**Summary Of The Review:**


Overall the paper is a direct adaption of MIMO to vision transformers. However, the design space is not thoroughly studied, and current results are only on small-scale dataset and small models, raising the concern regarding if current design is optimal and can adapt to other large models.

---

> ### Author Response · Authors · 2022-11-10
> **Response to reviewer jCdb (round 1)**
>
> Thank you for your feedback and comments.
>
> As noted in our general comments, we are very concerned by a **fundamental misunderstanding** in the **motivation** of this paper. **This paper sets out to address a core issue in the existing MIMO frameworks (lack of feature sharing between subnetworks)** as stated in the title, introduction and throughout the paper. MixViT achieves this by considering a new framework on ViTs and proposing a number of adaptations. Incidentally, experimental results (Sec. 4.4) show our MixViT is the only working MIMO framework on ViTs. As such, the primary goal of the paper **is absolutely not** to adapt MIMO models to ViTs. Rather, it is a step we take towards our goal of feature mutualization in MIMO models.
>
> It is therefore surprising to us that your review makes no mention of feature sharing/mutualization at all.
>
> Regarding your comments:
>
> **1) The paper introduces one way of bring MIMO to vision transformers, which is quite straightforward and novelty-limited, Source attribution is a direct analogy of using different input projection in existing MIMO. But there exist many other natural designs that can potentially work for vision transformers. The study space in the paper is not enough to show this proposed way is better. i.e. important baselines are missing. For example, use different patch embedding for different inputs (similar to different project for inputs in MixMo paper). Again, this specific design of late source embedding is not thoroughly studied. Early source embedding with carefully designs comparison is missing. Or middle source embedding, etc, this design should be carefully studied.**
>
> As discussed, the paper is not about bringing MIMO to ViTs, but about solving the feature sharing issue in MIMO models. Designing a MIMO ViT is a direct response to this problem, not the main point. As such, we do not think exploring the design space so thoroughly is necessary (it would be different if the paper were indeed simply about designing a MIMO ViT).
>
> We do study a few of the baselines you require however. Table 6 shows using different patch embeddings like in CutMixMo leads to very poor performance on ViTs (77.0% vs. 82.4% MixViT). Using source embeddings at the input level in MixViT (Appendix D) leads to slightly worse performance (82.2% vs 82.4%) and comes at the cost of a much larger computational overhead (x2 vs. x1.1).
>
> **2) Experiments are mainly conducted on small-scale datasets, ability to adapt to other large-scale datasets and large models remain unknown.**
>
> We do provide experiments on ImageNet(-1k) which is the gold standard large-scale benchmark.
>
> Additionally, we would like to point out small datasets are a typical pain point for Vision Transformer and have been the target of specialized study (Lee et al., 2021; Liu et al., 2021a). As such, we feel it is noteworthy that MixViT helps train particularly strong Vision Transformers on such small datasets.
>
> **3) Table 3 shows CaiT S-24 and M-24 results are 82.3 and 82.7. However, table 3 in CaiT paper shows s-24, m-24 results are 82.7 and 83.4 respectively. Can authors check the numbers here? And provides some large model (CaIT-M, or other large popular ViT variants) results would be better.**
>
> We are reporting reproduced results on our own settings (as we have done for all comparisons between the single-input/output baseline and MixViT for comparability). Our setting is slightly different than  CaiT (shorter training, some other differences). For logistical reasons it is difficult for us to run many experiments on ImageNet, which is both why the presented results could not be tuned and we cannot provide results on even larger models.

---

### Official Review · Reviewer_GaDC · 2022-10-25

**Confidence:** 4
**Correctness:** 2
**Technical Novelty And Significance:** 2
**Empirical Novelty And Significance:** 2
**Recommendation:** 3

**Clarity, Quality, Novelty And Reproducibility:**

- The motivation is clear and the method is technically sound, but the technical contribution is somewhat limited as discussed above.
- The writing is generally clear.

**Strength And Weaknesses:**

Strength
- Consistent improvements are achieved on the task of image classification.

Weaknesses
- The technical contribution seems limited. The input mixing for training is simply adopted from CutMix, and the source attribution appears to be quite straightforward.
- The model needs to run multiple times for the inference, causing a significant increase in the computational cost.
- The model is claimed to be multi-input and multi-output. However, it only supports two inputs and outputs, and the model needs to be changed for a different number of inputs, and it appears to be non-trivial to extend this model for more than two inputs and outputs. Hence, the model is not scalable and flexible for a different number of inputs and outputs.
- As this is an architecture paper, it would be more convincing to also validate the effectiveness on other tasks like segmentation and detection, for which the model seems not directly applicable.

**Summary Of The Paper:**

This paper introduces a transformer-based MIMO framework, called MixViT. At training time, two subsets of tokens from two images are taken as input and the model is tasked to predict labels for the two input images. At test time, the same input is passed to the model multiple times with different sourced tokens. The features in the earlier layers are shared while two separate encoders, called source attribution, are introduced in the later layer for the input-dependent feature extraction. In this way, the model is expected to achieve feature sharing and separation for multi-prediction. Improved results are obtained on image classification tasks compared with the single input counterparts and baselines.

**Summary Of The Review:**

Overall, I think the proposed method sounds interesting and good performance is also achieved. However, the technical contribution is somewhat limited as discussed above, and the weaknesses overwhelm the strength. Hence, I lean to recommend rejection based on its current shape.

---

> ### Author Response · Authors · 2022-11-10
> **Response to reviewer GaDC (round 1)**
>
> Thank you for your feedback and comments.
>
> As noted in our general comments, we are very concerned by a **fundamental misunderstanding** in the **motivation** of this paper. This paper sets out to **address a core issue in the existing MIMO frameworks (lack of feature sharing between subnetworks)** as stated in the title, introduction and throughout the paper. MixViT achieves this by considering a new framework on ViTs and proposing a number of adaptations. Incidentally, experimental results (Sec. 4.4) show our MixViT is the only working MIMO framework on ViTs. As such, the primary goal of the paper **is absolutely not** to adapt MIMO models to ViTs. Rather, it is a step we take towards our goal of feature mutualization in MIMO models.
>
> Feature mutualization is therefore not an afterthought or side-effect of the method. As a matter of fact, the entirety of Sec. 4.3 (as well as Appendix B) is dedicated to verifying MixViT properly shares features and showing how this benefits the model.
>
> Regarding your comments:
>
> **1) The technical contribution seems limited. The input mixing for training is simply adopted from CutMix, and the source attribution appears to be quite straightforward.**
>
> Evaluating the significance of the technical contribution is of course left to your appreciation. We would however like to point out three aspects we feel are missing from your evaluation:
> - We propose to use Vision Transformers to solve the feature sharing issues of traditional MIMO CNNs.
> - We completely change the traditional MIMO structure (which is made possible by source attribution).
> - Our new structure facilitates a new "parallel" inference scheme at little cost that appears necessary for MIMO ViTs (Sec. 4.4).
>
> We would also like to point out we make absolutely no claim that input cut-mixing is a contribution of ours (it is not). We slightly alter the traditional formulation to accomodate MIMO ViTs, which is why we take the time to describe the procedure in Sec. 2.
>
> **2) The model needs to run multiple times for the inference, causing a significant increase in the computational cost.**
>
> This is not the case. There is a slight increase in computational time as discussed in Sec. 2.2 and 2.4, but there is no need for multiple inference. If we had not altered the traditional MIMO structure, our parallel inference would indeed require multiple passes at inference.
>
> **3) The model is claimed to be multi-input and multi-output. However, it only supports two inputs and outputs, and the model needs to be changed for a different number of inputs, and it appears to be non-trivial to extend this model for more than two inputs and outputs. Hence, the model is not scalable and flexible for a different number of inputs and outputs.**
>
> MIMO models in general (a well-established family of models in the literature (Havasi et al., 2021; Rame et al., 2021; Sun et al., 2022; Cygert et al., 2022) suffer from this problem. In fact, this problem stems from the lack of feature sharing between subnetworks which is the core motivation of this work. Appendix B shows MixViT deals much better with more than 2 inputs/outputs then MIMO CNNs, though further research is required into input mixing schemes to truly benefit from more than 2 subnetworks.
>
> **4) As this is an architecture paper, it would be more convincing to also validate the effectiveness on other tasks like segmentation and detection, for which the model seems not directly applicable.**
>
> This is not an architecture paper. MixViT is not a new model, rather a MIMO framework to improve Vision Transformer models. The central question here is solving the feature mutualization issue in ViTs.
>
> While MIMO models are not traditionally applied to tasks other than classification, it is worth noting they have successfully been applied to object detection by Cygert et al. (2022) recently (and seem to particularly suffer from the lack of feature sharing in this case).
>
> (Cygert et al., 2022) Sebastian Cygert, Andrzej Czyzewski. Robust Object Detection with Multi-input Multi-output Faster R-CNN. In International Conference on Image Analysis and Processing, 2022.

---

### Official Review · Reviewer_6pnV · 2022-10-26

**Confidence:** 4
**Correctness:** 3
**Technical Novelty And Significance:** 3
**Empirical Novelty And Significance:** 2
**Recommendation:** 5

**Clarity, Quality, Novelty And Reproducibility:**

The paper uses certain phrases without clear definition, e.g. in "MixViT modifies the traditional MIMO structure to take full advantage of ViTs’ propensity to mutualize features between subnetworks while still retaining the advantage of training distinct predictions", it is not clear exactly what mutualize features mean. Can you show an example?

It seems MIMO networks using vision transformer is new.


**Strength And Weaknesses:**

Strength
1. This paper proposes MixViT, the first MIMO framework using vision transformers which takes advantage of ViTs’ inherent mechanisms to share features between subnetworks.

2. The paper shows that MixViT can have significant gains across multiple architectures (ConViT, CaiT) and datasets (CIFAR, TinyImageNet, ImageNet-100, and ImageNet-1k).

Weaknesses
1. It is not clear how MIMO training helps representation learning. The paper lacks analysis on this. Does MIMO training improve hard example classification accuracy? Does it better cluster images into different clusters? Does most of the gains come from input mixing such as CutMix?

2. The evaluation is on relatively small datasets. The paper should evaluate on larger datasets such as ImageNet.

**Summary Of The Paper:**

Similar to MIMO in wireless communication, multi-input multi-output training improves network performance by optimizing multiple subnetworks simultaneously. Previous MIMO network architecture uses CNN. This paper proposes MixViT, the first MIMO framework for vision transformers which takes advantage of ViTs’ inherent mechanisms to share features between subnetworks. Unlike MIMO CNN, MixViT only separates subnetworks in the last layers leveraging a source attribution that ties tokens to specific subnetworks. The paper shows that MixViT can have significant gains across multiple architectures (ConViT, CaiT) and datasets (CIFAR, TinyImageNet, ImageNet-100, and ImageNet-1k).

**Summary Of The Review:**

MIMO networks with vision transformer seems to novel. However, the paper does not provide deeper analysis on how it helps representation learning, besides the data augmentation benefits from input mixing, e.g. CutMix. The evaluation is also inadequate as it does not provide results on large datasets such as ImageNet.

---

> ### Author Response · Authors · 2022-11-10
> **Response to reviewer 6pnV (round 1)**
>
> Thank you for your valuable feedback and comments.
>
> While you do identify the core motivation of the paper, we would like to restate it here given the issues encountered by the other reviewers: MixViT addresses a core issue of existing MIMO frameworks (lack of feature sharing/mutualization between subnetworks) which causes a number of issues (need for wide models, difficulty in scaling to more than 2 subnetworks). Creating a MIMO ViT framework is therefore a proposed solution to this core issue, rather than the main goal of this paper.
>
> Regarding your two questions/issues regarding the paper:
>
> **1) It is not clear how MIMO training helps representation learning. The paper lacks analysis on this. Does MIMO training improve hard example classification accuracy? Does it better cluster images into different clusters? Does most of the gains come from input mixing such as CutMix?**
>
> **The paper uses certain phrases without clear definition, e.g. in "MixViT modifies the traditional MIMO structure to take full advantage of ViTs’ propensity to mutualize features between subnetworks while still retaining the advantage of training distinct predictions", it is not clear exactly what mutualize features mean. Can you show an example?**
>
> It seems to us the issue here is a difficulty in understanding what we mean by "feature mutualization/sharing" between subnetworks. We will revise the paper to clarify this once we converge on a satisfactory explanation in this discussion.
>
> What typically occurs in MIMO CNNs (as studied at length by Sun et al. (2022)) is that 2 subnetworks within a MIMO model "do not share features". What does this mean? Let us consider a final feature vector f=(f_0, ..., f_3) and 2 classifiers d_0 and d_1. Then, f_0 is only going to influence the decision of one of the classifiers ("classical MIMO problem" in Fig. 1). Similarly, f_0 is only going to be influenced by one of the inputs (can be observed by freezing one input and looking at variance over the dataset on the other input). The same behavior emerges throughout MIMO models (in intermediary feature maps). To the contrary, MixViT trains subnetworks such that f_0 is used by both classifiers and depends on both inputs.
>
> Why is this important? This lack of feature sharing restricts MIMO training to large base models and prevents training more than 2 subnetworks. Moreover, this means features cannot benefit from being used by two subnetworks at once. In theory, a feature being useful to describing two different inputs should be more generally useful than a feature used to describe a single input.
>
> We show in Sec. 4.3.1 that our learned features are indeed mostly shared between subnetworks (close to the diagonal on the plot) whereas traditional MIMO models are not (close to the axis). Sec. 4.3 additionally demonstrates MixViT is much less reliant on large base models than MIMO CNNs and seems to benefit from some sort of implicit regularization due to the shared features. Appendix B shows MixViT behaves much better with more than 2 subnetworks compared to traditional MIMO CNNs.
>
> Regarding your last point on the gains from cutmixing inputs, it is indeed a fair question. With a single-input/output baseline at 79.5% accuracy, Tab. 6 shows using only Input Cut-mixing in ViTs (+ our parallel inference, otherwise the performance is very poor) yields a performance of 80.1% accuracy against 82.4% for MixViT. As such, the gains do not seem to come solely from cutmixing.
>
> **2) The evaluation is on relatively small datasets. The paper should evaluate on larger datasets such as ImageNet.**
>
> We do propose experiments on ImageNet in Sec. 4.2: ImageNet-1k corresponds to the seminal 2012 dataset usually used in the literature. We only added "-1k" to differentiate from ImageNet-100 in our results, but are open to changing the name to ImageNet if you feel it better.
>
> Additionally, we would like to point out small datasets are a typical pain point for Vision Transformer and have been the target of specialized study (Lee et al., 2021; Liu et al., 2021a). As such, we feel it is noteworthy that MixViT helps train particularly strong Vision Transformers on such small datasets.

---

### Author Response · Authors · 2022-11-10
**General comments from the authors to the reviewers (round 1)**

We thank all reviewers for their insightful comments and for the time invested in reviewing our work. We have addressed all reviewers individually, but there is one point we feel important to stress.

We are quite concerned by a **strong misunderstanding** regarding the **motivation** of this paper: **This paper sets out to address a core issue in the existing MIMO frameworks (lack of feature sharing between subnetworks)** as stated in the title, introduction and throughout the paper. MixViT achieves this by considering a new framework on ViTs and proposing a number of adaptations. Incidentally, experimental results (Sec. 4.4) show our MixViT is the only working MIMO framework on ViTs. As such, the primary goal of the paper **absolutely is not** to adapt MIMO models to ViTs. Rather, it is a step we take towards our goal of feature mutualization in MIMO models.

Given the widespread confusion around this issue (3 reviewers out of 4), the writing of the paper might need some adjustments on the matter. As such, we would like to work with reviewers to understand how to make this clearer in the time of this discussion.

---

### Decision · Program_Chairs · 2023-01-20

**Decision:**

Reject

**Justification For Why Not Higher Score:**

The rating of the paper could be improved by the presentation quality and improving the main experimental setting in Table 1.

**Justification For Why Not Lower Score:**

N/A

**Metareview: Summary, Strengths And Weaknesses:**

Summary: The authors propose a way to train a practical MIMO method that has feature sharing. They use the ViT architecture and image classification benchmarks for their experiments. The main idea is to mix the tokens in the input, use a shared encoder (that allows for feature sharing), and use source attributions in the final layers to perform classification.

Strengths:
- Clever use to ViT model's ability to share features.
- MixViT shows improvements on many small-scale image classification benchmarks (Table 1).

Weaknesses:
- A majority of the reviewers found issues with the writing and presentation of the paper, and remain unconvinced even after the author response. On reading the paper, the AC agrees that the important details only become clear after reading the author response. We would encourage the authors to improve the presentation
- Training recipes in Table 1 are not controlled for, which makes the comparison unclear.